# A Standardized Operative Protocol for Fixation of Proximal Humeral Fractures Using a Locking Plate to Minimize Surgery-Related Complications

**DOI:** 10.3390/jcm12031216

**Published:** 2023-02-03

**Authors:** Sebastian Kwisda, Jan-Philipp Imiolczyk, Tankred Imiolczyk, Magdalena Werth, Markus Scheibel

**Affiliations:** 1Department of Shoulder and Elbow Surgery, Schulthess Clinic, 8008 Zurich, Switzerland; 2Center for Musculoskeletal Surgery, Charité-Universitaetsmedizin Berlin, 13353 Berlin, Germany; 3Department of Mathematics, University of Mannheim, 68131 Mannheim, Germany

**Keywords:** proximal humerus fracture, angular stable plating, standardized surgical protocol, adverse events, complication, biological, ORIF, PHILOS

## Abstract

The current literature suggests that up to 55% of complications after plate osteosynthesis treatment for patients with proximal humerus fractures are attributed to the surgical procedure. The hypothesis of this study was that a standardized surgical protocol would minimize surgery-related adverse events. This prospective cohort study included 50 patients with a mean age of 63.2 (range 28–92) years treated by one single surgeon using a previously published standardized surgical protocol. Clinical and radiological follow-up examinations were conducted for up to 24 months using Constant–Murley Score (CS), Subjective Shoulder Value (SSV) and radiographs in true anteroposterior, axial and y-view. Finally, CS was 73.9 (standard deviation [SD]: 14.0) points (89% compared to the uninjured shoulder), and SSV was 83.3% (SD: 16.7) at two years of follow-up. Postoperative radiologic evaluation revealed no primary surgical-related or soft-tissue-related complications (0%). The main complications were secondary, biological complications (20%), largely represented by avascular necrosis (8%). Eight patients underwent revision surgery, mainly for implant removal. In addition, a total of four patients were revised using a hemiarthroplasty (n = 2), reverse shoulder arthroplasty (n = 1) or re-osteosynthesis. The use of our standardized surgical technique on proximal humerus fractures improves fixation with regard to primary stability and prevents primary, surgical-technique-related complications. The subjective grading of a high level of difficulty surgery was associated with more complications.

## 1. Introduction

Proximal humerus fractures (PHF) account for approximately 45% of all humeral fractures [1,2]. It is already the third most common fracture (10%) in patients over 65 years of age [3,4,5]. In an increasing elderly population, its incidence is expected to increase even further [6,7,8,9,10]. Due to osteopenia, the number of complex fracture patterns also rises. In young patients (under 50 years), PHF are more seldom; however, they are mainly caused by high-energy trauma in male patients around 40 years [11,12,13,14]. Unstable and dislocated fractures regularly require surgical treatment and often pose a challenge to surgeons [15,16,17,18,19,20,21]. There is no consensus on the optimal treatment of PHF as of yet.

Open reduction and internal fixation (ORIF) with a locking plate is an accepted, commonly used and widely available method [22,23,24,25,26,27,28,29,30,31,32]. The angular stable system, with its high rigidity, is specifically designed to address complex fracture patterns, especially in patients with poor bone quality. Even though long-term results (10 years) suggest that this technique provides good to excellent clinical results, high complication rates of up to 34% have been reported [18,22,25,31,33,34,35,36,37]. One can distinguish between primary and secondary complications: primary complications include surgical, technique-related and soft tissue-related complications; secondary complications consist of “biological” and implant-related complications. While “biological” complications like avascular necrosis (AVN) with secondary collapse depend on the fracture pattern and its impact on humerus head ischemia, up to 55% of complications are directly associated with the surgical procedure itself and implantation technique [23,31,32,38,39]. These surgical, technique-related complications include primary screw perforation, plate impingement and malreduction, which are usually visible on the postoperative radiographic examination.

A conclusive evaluation of overall complication rates and clinical results is difficult owing to the plethora of different surgical techniques regarding approach, fracture repositioning and retention, as well as the inconsistent use of allografts and suture cerclages due to personal preferences.

A standardized approach for anatomical reduction, retention and fixation using a locking plate osteosynthesis was published in 2018 by Minkus et al. [40]. This reproducible technique allows for anatomical reduction and fixation with high primary stability. This standardized, surgical step-by-step protocol has been used for all PHF regardless of fracture type, degree of displacement or tuberosity comminution.

The purpose of this study was to evaluate clinical and radiological results as well as adverse events of this standardized approach within a two-year time period. According to our hypothesis, the use of this protocol would minimize the number of primary surgery-related adverse events with good to excellent clinical results regardless of fracture type. Furthermore, our aim was to assess the outcome of simultaneous biceps treatment (tenotomy vs. tenodesis) in this specific patient cohort.

## 2. Materials and Methods

### 2.1. Study Population

This trial was approved by the ethics committee of Charité-Universitaetsmedizin Berlin. Informed, written consent was obtained from all patients. Between January 2010 and October 2012, 50 patients (35 women, 15 men) who had suffered a proximal humerus fracture (PHF) and were in need of surgical intervention were included in this prospective, non-randomized, single-center trial. The mean age at the time of the trauma was 63.2 years (range 28–92). Only patients over 18 years were included. The included fracture types were classified using AO/OTA Fracture and Dislocation system. Every fracture was surgically treated by ORIF using a PHILOS plate (“Proximal Humerus Internal Locking System”; Synthes DePuy GmbH, Oberdorf, Switzerland).

### 2.2. Surgical Procedure and Postoperative Protocol

The surgery was performed by the previously published technique by Minkus et al. [40]. The surgery was performed in a beach chair position on a translucent table (Figure 1a), with the arm in a floppy position on a side table under general anesthesia and interscalene block. Each patient received intravenous antibiotics within 30 min of surgery. At the beginning of the procedure, the fracture was re-evaluated in a true a/p (Figure 1b) and axillary view (Figure 1c,d) using an image (radiographic) intensifier. A standard deltopectoral approach with an incision directly over the anterior portion of the deltoid muscle was used. Subsequently, the fracture was exposed, the hematoma was evacuated, the bursa was partially removed and a deltoid retraction hook (Innomed, Savannah, GA, USA) was inserted underneath the deltoid muscle for maximum exposure of the fracture lines between shaft, head and the greater tuberosity (Figure 1e). Depending on the age of the patient (taking into consideration his/her cosmetic demands), either a tenotomy or tenodesis of the long head of the biceps tendon was performed.

To reduce tension of the deltoid muscle and to increase exposure, the arm was abducted. The tuberosity fragments were armed with Fiber Wire No. 5 (Arthrex, Naples, FL, USA) at the tendinous–osseus junction (Figure 1f).

The posterior rotator cuff was armed with one suture cerclage each in the infraspinatus and one in the teres minor tendon to primarily help with anatomical reduction and secondarily neutralize the posteromedial pull of the posterior rotator cuff. In addition, Fiber Wire sutures were placed through the subscapularis tendon. Three percutaneous 2.0 mm K-wires were drilled retrograde into the humeral shaft towards the fracture line between head and shaft (i.e., waiting position) (Figure 1g,h).

Following this, the head fragment was anatomically reduced (Figure 1i), especially the integrity of the medial buttress, using an image intensifier, and the K-wires were advanced into the humeral head until they reached the subchondral bone plate to retain the humeral head in position (Figure 1j). Next, the tuberosity fragments were reduced. Impacted fractures often entail a loss of bone stock at the lateral aspect of the humeral head. In these cases, one to two allografts were shaped and impacted into the fracture site to allow anatomical reduction of the tuberosity fragment and, hence, prevent a valgic displacement of the humeral head (Figure 1k,l). Depending on the age and functional and cosmetic demands of the patient, either a biceps tenotomy or tenodesis was performed (Figure 1m). Following successful visualization of anatomical reduction, the Fiber Wire sutures of the anterior and posterior cuff were tightened together (Figure 1n). The angular locking plate (PHILOS) was then introduced (Figure 1o), and the position of the plate was controlled using an image intensifier in true anterior–posterior view to avoid subacromial impingement. The plate was temporarily fixed with 1.6 mm K-wires proximally and distally, and the position was re-evaluated in an axillary view (Figure 1p). Before locking screws were inserted, the plate was temporarily fixed to the proximal humerus using a standard 3.5 mm cortical screw that was inserted into the center of the long shaft hole of the plate. Once the screw was inserted, the K-wires were removed. After pre-drilling (Figure 1q), the screws were meticulously placed from proximally to distally (Figure 1r). Two bicortical locking screws were inserted in the two most inferior plate holes. If support for the medial buttress was needed, one or two calcar screws were used to provide additional stability. Finally, the 3.5 mm cortical screw was exchanged for a locking screw. In very osteoporotic bone, drilling was performed under fluoroscopy since penetration of the drill into the joint would have been difficult to notice. The correct subchondral position of these screws was double checked by means of multi-plane fluoroscopy to avoid penetration of the screws into the glenohumeral joint. Thereafter, the three temporary K-wires were retrieved and the final construct was re-evaluated in internal and external rotation as well as abduction (Figure 1s–u) to evaluate plate impingement and stability.

Postoperatively, the arm was functionally immobilized in a neutral rotational brace for 4 weeks, and passive range of motion exercises were initiated on day two after surgery. Active range of motion was started after 4 weeks.

### 2.3. Intraoperative Assessment

Surgeons also rated bone quality on a scale from 1 (excellent) to 5 (poor) and difficulty of surgery (easy, moderate, difficult).

### 2.4. Clinical Examinations

Patients were evaluated according to the hospital’s standard regime, i.e., after 6 and 12 weeks as well as 6, 12 and 24 months after surgery. Postoperative evaluation was undertaken by the same examiner in order to avoid inter-observer variability. A physical examination, including active and passive range of motion (ROM) of forward elevation, abduction, internal and external rotation of the injured shoulder, was performed at each follow-up visit. The numeric analogue scale (NAS) and Constant–Murley Score (CS) of the injured and uninjured shoulder were determined at the 6, 12 and 24 months examination [41,42]. At 1 and 2 years after surgery, patients were additionally evaluated by means of the Subjective Shoulder Value (SSV) and the Long Head of Biceps-Score (LHB-Score) [43,44].

### 2.5. Radiographic Evaluation

Radiographs were routinely obtained with a true anterior–posterior, a Y- and axillary view directly postoperative, at 6 weeks, 12 weeks, 6 months, 1 and 2 years after surgery using a standardized evaluation protocol. Radiological follow-up after 6 and 12 weeks was primarily used to detect surgical technique-related complications. Evaluation at later time points investigated fracture healing (union, delayed or non-union), implant failure (breakage, loosening, displacement), screw cut-out, avascular necrosis and loss of reduction. In cases of complications or after revision surgery, radiographic follow-up continued until the end of treatment.

### 2.6. Adverse Events

All intraoperative and postoperative surgery and implant-related complications were documented as adverse events within the follow-up period of 24 months.

We have distinguished between primary and secondary complications. Primary complications include surgical technique related (i.e., primary screw perforation, plate impingement, malreduction) and soft tissue-related complications (e.g., superficial or deep wound infection, neurological lesions). Secondary complications were divided into implant related (e.g., screw loosening, implant breakage) and “biological” as in fracture or bone related (i.e., AVN, loss of reduction with or without screw perforation, non-union).

### 2.7. Statistical Analysis

Data analysis was performed with SPSS 25.0 (SPSS Inc, Chicago, IL, USA). Patients’ characteristics were summarized using descriptive statistics, including means and ranges. We used a Mann–Whitney U Test and a paired *t*-test to detect significant differences in scores and ROM during the various follow-up examinations as well as for subgroup analysis when comparing patients with LHB-tenodesis versus tenotomy. For comparison analysis regarding three groups or more (difficulty of the surgeon or bone quality) ANOVA was used. Statistical level of significance was set at 0.05.

## 3. Results

Fifty patients who had suffered a PHF and were treated in the described standardized fashion by one single surgeon (M.S.) at our center were identified for this analysis. Dropout occurred in 18% of patients: six patients did not wish to participate in any follow-up examination after surgery, while some dropped out after showing the first satisfactory results after 6 (n = 2) or 12 weeks (n = 1). This has left a total of 41 patients who were available for the one (n = 40) or two (n = 38) year follow-up examination or both (n = 35). In four cases, the initial locking plate was revised due to complications after 12 months, and they were excluded from the 24-month follow-up examination (n = 34). Those patients are further outlined below.

Fracture classifications, as well as surgical specifications, are listed in Table 1. The duration of surgery was, on average, 116 (range: 80–180; SD: 28) minutes.

### 3.1. Clinical Results

Active range of motion had significantly improved from three to six months (*p* < 0.001) and from 12 to 24 months (*p* < 0.05). Between six and 12 months after surgery, the active range of motion had also improved but not significantly for any of the tested motions (*p* > 0.05).

As shown in Figure 2, there has been a steady improvement in CS, forward flexion and external rotation from the six months to the one and two-year follow-up examinations.

The CS improved significantly between each follow-up examination from 6 to 12 and as well as from one- to two-year follow-up (*p* < 0.001). As expected, the CS of the contralateral shoulder remained almost constant through the postoperative follow-up (i.e., 82.5 ± 5.5 at 12 weeks and 84.2 ± 4.8 two years after surgery).

Differences in SSV between one and two years were not significant (*p* > 0.05). Clinical results after six, twelve and 24 months postoperatively are displayed in Table 2. 

### 3.2. LHB Tenodesis vs. Tenotomy

The mean age of patients in the tenodesis group was 55.6 ± 12.7 years and 70.3 ± 13.0 years in the tenotomy group. Forty out of the 50 patients who were present at the one-year follow-up had either been treated with tenodesis (20 patients, 50%) or tenotomy (20 patients, 50%). Patients in the tenodesis group had an LHB-Score of 93.2 ± 8.0 (range 72–100), and patients after tenotomy had 95.1 ± 5.0 (range 82–100) one year after surgery. There was no significant difference between both procedures (*p* = 0.7). Scores for the contralateral shoulder were identical in both groups (99.9 ± 0.2). Two years after surgery, 16 patients who had received a tenodesis and 18 with tenotomy were present. There was no significant difference (*p* = 0.46) between the tenodesis 93.4 ± 7.8 (range 72–100) and the tenotomy 97.6 ± 3.3 (range 92–100). Improvements within either group from the first to the second evaluation were not significant.

There were no significant differences concerning ROM, CS or SSV with regard to either gender or subjective bone quality. There was one significant difference with regard to the subjective difficulty of surgery.

### 3.3. Complications

Complications occurred in a total of 10 patients, and 8 needed a re-operation (Table 3). The postoperative evaluation did not show any primary screw cut-out, malpositioning or subacromial impingement of the plate. There were no (0%) primary, surgical-technique-related or soft-tissue-related complication in our cohort.

Secondary biological complications, as in partial or complete AVN (n = 5), secondary screw cut-out (n = 2) pseudarthrosis (n = 1) or secondary displacement of the greater tuberosity (n = 1), were 80% of all complications. Every case of AVN resulted in secondary surgery with the need for implant removal; additionally, in two cases, a revision to hemiarthroplasty and one revision to reverse shoulder arthroplasty was necessary. Both patients with secondary screw-cut were reoperated, and the implant was removed with no further intervention. In the case of secondary GT displacement, no re-operation was necessary. The patient with pseudoarthrosis was treated with a re-osteosynthesis after 12 months.

Secondary implant-related complications included partial plate loosening (n = 1), which did not need any reintervention.

During the follow-up period, three patients needed conversion to hemiarthroplasty (n = 2; 4%) or reverse shoulder arthroplasty (n = 1; 2%).

Out of 17 treated so-called “A-fractures”, three (18%) patients reported complication, while four “B-fractures” (out of 15; 27%) and three “C-fractures” (out of 18; 17%) showed complication in the time two years after surgery. Only three patients were rated as having excellent bone quality (grade 1), showing 0% complication; 21 patients were with good bone quality (grade 2; 24% complication); 19 patients were with so-and-so quality (grade 3; 26% complication) and seven patients werewith poor bone quality (grade 4), showing no postoperative complication. There were no patients being intraoperatively evaluated with the worst grade of bone quality (grade 5). Twenty-seven surgeries were rated as easy, nineteen as moderate and four as difficult. Easy operations were subjected to complication in 7% of cases in our study, while surgeries of moderate difficulty showed higher complication rates (21%). Nevertheless, every difficult operation (100%) did result in a complication with a secondary screw cut-out needing implant removal.

## 4. Discussion

Our study shows comparable clinical and radiological results with a complication rate of 20% with this standardized protocol. There were no primary surgical-related or soft-tissue-related complications. All complications were secondary complications in our cohort. Clinical results were satisfying, with a mean absolute CS of 74 points, a relative CS of 89% compared to the contralateral shoulder and an SSV of 83% after two years post-surgery. Patents gained most of the function in the first six months, with statistically significant increases in range of motion and CS from three to six months. There was a consistent increase in function from each follow-up examination onwards to the next; nevertheless, it did not reach clinical or statistical significance.

Out of 10 patients who developed a complication during follow-up, 8 (16%) needed reoperation, with 8% needing either re-osteosynthesis or conversion to shoulder arthroplasty. Nevertheless, in our cohort, no complication could be attributed to the initial surgical procedure itself. In accordance with this, there was no case of primary malposition of the plate or primary screw cut out.

Consistent with published data, AVN was the most common complication and arose in 10% of our patients [32,45,46,47]. All cases of AVN were diagnosed between 6 and 12 months after surgery. However, no additional cases of AVN were diagnosed past the 1-year evaluation. This concurs with the current literature that AVN remains absent after one year [48].

Considering the three intraoperative measures, fracture type and subjective bone quality did not play a significant prognostic factor for a greater risk of complication. However, regarding the subjective difficulty of surgery, our data show that difficult operations are prone to complications. All subjectively difficult characterized surgeries (4/4; fracture type B2, B3, C2, C3) have led to secondary screw cut-out, although the subjective bone quality was rated moderately (2 × Grade 2 and 2 × Grade 3).

Our study showed no significant difference between tenodesis and tenotomy regarding the outcome. This shows that the indication for either should depend on age, functional requirements as well as patients’ preferences and cosmetic demand. As patients in the tenotomy group were, on average, 15 years older, they did not show any functional or cosmetic impairments resulting in such significant differences in the LHB score. If a proper patient selection is performed, tenotomy or tenodesis of the LHB leads to the same results after open reduction and internal fixation, as shown previously by Kerschbaum et al. [49].

Open reduction and internal fixation with a locking plate is a commonly used technique that is widely available and can provide good to excellent clinical results comparable to ours [22,23,24,25,26,27,28,29,30,31,32,50]. However, high complication rates have been reported [18,22,25,31,33,34,35,36,37,46,47,51] in the literature study. This has resulted in a therapeutic shift towards implantation of RSA for elderly patients that show slightly worse function but avoid unnecessary secondary surgery after complications [37,46,52]. Nevertheless, in the literature, there has never been one standardized, structured surgical technique but rather implants or fracture morphologies. Our hypothesis was that this standardized step-by-step technique presents advantages and reduces complications.

This technique enables perfect reduction of the medial calcar through temporary percutaneous K-wire fracture stabilization. Furthermore, the use of allografts results in additional stability and anatomic bone stock augmentation. The Fiber Wire tensioning allows perfect reduction through intra-tuberosity cerclages, which should enhance tuberosity healing regardless of fracture pattern or tuberosity comminution. The step-by-step surgical technique enables anatomical retention and fixation of PHF using a locking plate.

Current meta-analyses have shown that patients undergoing ORIF provide better clinical outcomes and range of motion compared to HA and RSA; however, complication rates are in favor of treatment with RSA [37,52]. With RSA being a viable treatment option, especially for elderly or multimorbid patients, complication rates for ORIF have decreased in the last decades to roundabout 20% and revision rates of 10% [22,25,33,34,35,36,37,51,53,54,55]. Nevertheless, reconstruction should be the goal for young and active patients with proximal humerus fractures. Although elderly patients can profit greatly from RSA as fracture treatment [56,57,58,59], ORIF for reconstructable fractures with good bone stock should be considered when the medial calcar is intact because the old age of the patient itself is not associated with poorer outcome [46,50,60,61,62,63].

Most of the literature did not focus on a standardized protocol for retention and fixation and has published clinical and radiographic data without emphasizing their surgical technique, which begs the question of whether there was a standardized protocol or if the surgery was performed by the surgeons’ best skill and to his favorite preference. Considering the approach, the literature shows that there is no significant difference in functional outcome as opposed to the deltoid split [64,65]. To put our data into perspective, we have compared this to some of the literature that was multicentric or multi-surgeon protocol or both to increase the chance of there being no standardized surgical technique used for all patients.

One study that compared conservative treatment and ORIF performed by multiple surgeons for two-part PHF showed no significant differences for both groups, with a final CS of 68 (±3.2) points after two years for patients treated with a PHILOS plate [66]. In 9% (3 out of 33) of followed-up patients in the surgical group, two were subject to a secondary screw-cut out and one to a periprosthetic fracture after a fall. Conservative treatment for C-type fractures is associated with greater pain levels in the post-acute stage and with worse function in the long term [61].

A multicenter randomized control trial has shown favorable outcomes for not-severe displaced C2 fractures in patients between 65 and 85 years for RSA over ORIF [56]. Complications occurred in eleven patients (18%) (11/60; 9 patients with screw penetration) treated with ORIF; eight of those needed revision surgery, and four were converted to RSA.

This study has several limitations. Despite its prospective design, this study displays a relatively small sample size with follow-up rates of 80% after one and 76% after two years. These results fall within the range of published follow-up rates between 60% and 87% [67,68,69]. As all patients in our clinic were treated within this standardized protocol, there was no control group. Moreover, all patients were treated by one specialized senior shoulder surgeon.

## 5. Conclusions

The use of a standardized technique for open reduction and internal fixation of proximal humerus fractures improves fixation with regard to primary stability and prevents primary, surgical-technique-related complications. This step-by-step surgical technique allows anatomical retention and fixation of any PHF regardless of fracture pattern, grade of displacement or tuberosity comminution.

The difficulty of surgery is associated with higher rates of complication, whereas bone stock and fracture patterns did not. Patients with tenotomy and tenodesis presented with identical LHB-specific results after open reduction and internal fixation.

## Figures and Tables

**Figure 1 jcm-12-01216-f001:**
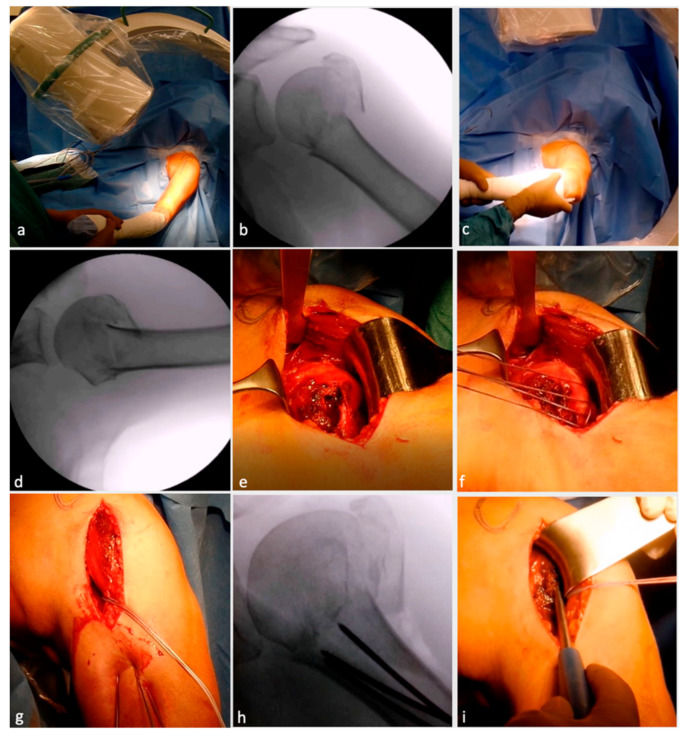
Surgical technique illustration of locking plate osteosynthesis with allograft augmentation. Beach chair positioning (**a**) with image intensifier placed so that true anteroposterior view (**b**) and axial views (**c**,**d**) are possible. Deltopectoral approach with fracture exposure (**e**). Arming of tuberosities (**f**). Percutaneous, retrograde drilling of K-wires (**g**) up to the fracture line (**h**). Reposition and reduction of head fragments (**i**) and K-wire advancing till subchondral bone (**j**). Osseous defect due to fracture visible (**k**) and augmention with allograft (**l**). Treatment of bicep tendon (**m**) followed by tuberosity refixation (**n**). Angular locking plate placed in situ (**o**) and fixed temporarily with K-wires (**p**). Drilling from top to bottom (**q**) followed by screw placement (**r**). Final evaluation for stability and impingement in situ (**s**) and under image intensifier in true anteroposterior (**t**) and axial view (**u**). (Reproduced, with modification, under Creative Commons Attribution 4.0 International. License [https://creativecommons.org/licenses/by/4.0/], from [40]).

**Figure 2 jcm-12-01216-f002:**
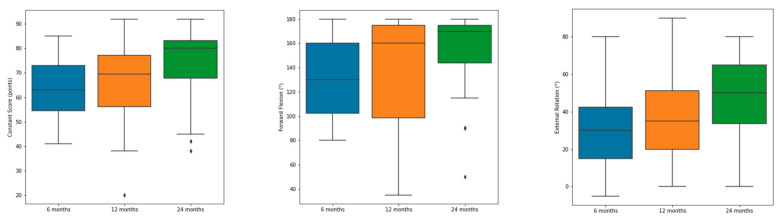
Improvement of absolute Constant–Murley Score, forward flexion and external rotation from 6 over 12 to 24 months. Minimum, maximum and average are displayed in the boxplot. Outlier are displayed as rhombus.

**Table 1 jcm-12-01216-t001:** Fracture classifications, operative characteristics and intraoperative assessment for all included patients (n = 50).

Operative Characteristics	n
AO/OTA Fracture and Dislocation Classification	
A2	4
A3	13
B1	7
B2	6
B3	2
C1	12
C2	5
C3	1
Additional treatment	
Fiber wire Cerclage GT	24
Fiber wire Cerclage LT	2
Fiber wire Cerclage GT + LT	9
Fiber wire cerclage around proximal humerus	3
No Fiber wire cerclage	12
Bone augmentation	20
Biceps tenodesis	24
Biceps tenotomy	26
Additional anterosuperior rotator cuff reconstruction	6
Rating of surgery (difficulty) (subjective)	
easy	27
moderate	19
difficult	4
Bone quality (1 = good; 5 = bad) (subjective)	
1	3
2	21
3	19
4	7
5	0

AO/OTA—AO classification according to Orthopedic Trauma Association; GT—greater tuberosity; LT—lesser tuberosity.

**Table 2 jcm-12-01216-t002:** Clinical results and mean of diefferences with regards to Constant–Murley Score, Subjective Shoulder value and SSV after 6 months, one and two years.

Follow-Up Examination	Forward Flexion (°)	Abduction	External Rotation (°)	Internal Rotation	Absolute CS (Points)	Relative CS * (%)	SSV (%)
6 months (n = 35)	132 (±32) (80–180)	118 (±38) (40–180)	31 (±20) (−5–80)	L2 (buttocks–T5)	62 (±13) (41–85)	82 (±10) (42–100)	-
12 months (n = 40)	138 (±42) (35–180)	129 (±45) (30–180)	38 (±26) (0–90)	T12 (buttocks-T3)	66 (±16) (20–92)	84 (±7) (52–100)	77 (±20) (10–100)
24 months (n = 34)	153 (±35) (50–180)	146 (±38) (55–180)	48 (±23) (0–80)	T10 (buttocks-T3)	74 (±14) (38–92)	89 (±8) (43–100)	83 (±17) (5–100)
Mean differences							
between 6 and 12 months (95% CI)	16 (3;30)	20 (7;33)	11 (2;19)	3 vertebra (2;4)	6 (1;11)	6 (3;10)	-
between 12 and 24 months (95% CI)	6 (1;12)	9 (2;15)	6 (0;11)	1 vertebra (1;1)	4 (2;6)	4 (2;7)	3 (−1–8)

CS—Constant–Murley-Score; SSV—Subjective Shoulder Value; CI—Confidence interval; L2—2nd lumbar vertebra; T5—5th thoracic vertebra; T12—12th thoracic vertebra; T3—3rd thoracic vertebra; T11—11th thoracic vertebra. * compared to the contralateral shoulder

**Table 3 jcm-12-01216-t003:** Patients with complication.

No.	Fracture Type	Difficulty of Surgery	Bone Quality *	Complication	1Y CMS (Points)	1Y SSV (%)	2Y CMS (Points)	2Y SSV (%)	Revision
1	C2	difficult	3	Secondary screw cut-out	68	60	-	-	Implant removal after 1Y, no 2 year FU
2	B2	moderate	2	Secondary screw cut-out	54	80	71	90	Implant removal
3	A3	moderate	3	Partial (distal) plate loosening	57	70	56	70	No revision surgery
4	A3	easy	2	Total AVN with secondary screw cut-out	41	60	37	60	Revision to reverse shoulder arthroplasty after 12 months
5	B3	difficult	2	Partial AVN with secondary screw cut-out	20	10	18	5	Revision to hemiarthroplasty after 12 months
6	C3	difficult	2	Partial AVN with secondary screw cut-out	78	75	83	65	Implant removal and arthroscopic capsulotomy
7	C1	easy	3	Secondary GT displacement and extraanatomical consolidation	49	50	-	-	No revision, no 2-year FU
8	B2	difficult	3	Partial AVN with secondary screw cut-out	68	90	68	80	Implant removal
9	A3	moderate	3	Partial AVN with secondary screw cut-out	39	65	60	90	Revision to hemiarthroplasty after 12 months
10	B2	moderate	2	Pseudoarthrosis	45	35	45	70	Implant removal and Re-ORIF after 12 months

AVN—avascular necrosis; GT—greater tuberosity; FU—follow-up; ORIF—open reduction and internal fixation. * bone quality from 1 (best) to 5 (worst).

## Data Availability

Not applicable.

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
