# Peer review of "A Standardized Operative Protocol for Fixation of Proximal Humeral Fractures Using a Locking Plate to Minimize Surgery-Related Complications"

_jcm, 2023, doi:10.3390/jcm12031216_

Round 1
Reviewer 1 Report
Thanks to the Editor for inviting me to review this exciting study. I congratulate the authors for this well-written and well-structured manuscript. The authors conducted a non-randomized, single-center prospective study to assess the clinical and radiological results of a standardized approach to proximal humerus fractures by Minkus et al. and related adverse events. By following this approach, the authors found satisfactory surgical and radiological outcomes (CS, SSV) and a significant decrease in primary complications at 2 years.
STRENGTHS
Optimal methodology.
Single-surgeon and standardized (surgical steps and implants) approach nature.
This standardized approach study represents a breakthrough in proximal humerus fracture management, improving the quality of patient care worldwide.
WEAKNESSES
The authors do not disclose how the sample size was defined.
It seems that no analysis was conducted on complications and fracture types, the surgery difficulty reported by the surgeon, and the bone quality of the patients with complications.
References seem outdated. Only four out of fifty-three were published in the last five years.
SECTION BY SECTION
Introduction:
Optimal.
Materials and methods:
Please, include biceps surgical treatment among the aims and outcomes of the study.
Please, clarify how the sample size was defined. Did the researcher conduct a previous or post-hoc power analysis?
Is the surgical technique exactly as described by Minkus et al.? Please, highlight the modifications regarding the original publication, if any.
Line 88. Please, correct “delta” to deltoid.
Results:
Line 192. Please, confirm that the total number of patients was 41 at this point. When doing the math seems 42.
Please add the mean difference and 95% confidence interval when presenting the clinical results.
Please, add the fracture types, the surgery difficulty reported by the surgeon, and the bone quality of the patients with complications, and conduct an analysis considering these factors. This is very important, as complications might be present in the fractures, surgeries with higher complexity degrees, or poor bone quality.
Discussion:
Please, start the discussion with the main findings of the study.
Please, consider restructuring the discussion following these comments:
a. Complication profiles within techniques might be related not only to the surgical approval (open/miniopen), but also to the fracture type and bone quality. However, the discussion on ORIF did not focus on these parameters.
b. Lines 306-307. “Nevertheless, a reconstruction should be the goal for young and active patients with proximal humerus fractures”. Please, expand on this idea. The study cohort has a mean age of 63.2 (range 28-92) years, and all the patients were surgically treated with the standardized approach regardless of their age and fracture type. Does that mean all adult patients should undergo ORIF as a primary procedure? Should age only be considered when addressing the biceps tendon?
c. According to the present study findings, what should be the role of shoulder arthroplasty?
Conclusion:
The conclusion seems fine but should be revised after addressing the previously mentioned concerns.
Title:
Please, consider rewording retention to reduction.
Abstract:
The abstract seems fine but should be revised after addressing the previously mentioned concerns.
Line 20. “… minimum of 24 months… ”
References:
References seem outdated. Only four out of fifty-three were published in the last five years. Please, consider its update according to the Editor’s advice.
Tables:
Table 3. Please, add the fracture types, the surgery difficulty reported by the surgeon, and the bone quality of the patients with complications.
Figures:
The figures are adequate. However, splitting the images of the surgical technique might improve its visualization.
Author Response
|
Comments |
Author responses |
Changes made |
|
Thanks to the Editor for inviting me to review this exciting study. I congratulate the authors for this well-written and well-structured manuscript. The authors conducted a non-randomized, single-center prospective study to assess the clinical and radiological results of a standardized approach to proximal humerus fractures by Minkus et al. and related adverse events. By following this approach, the authors found satisfactory surgical and radiological outcomes (CS, SSV) and a significant decrease in primary complications at 2 years.
|
Thank you very much for your feedback. We have tried to implement all your feedback accordingly. |
|
|
Optimal methodology. Single-surgeon and standardized (surgical steps and implants) approach nature. This standardized approach study represents a breakthrough in proximal humerus fracture management, improving the quality of patient care worldwide.
|
Thank you very much. |
|
|
The authors do not disclose how the sample size was defined.
|
We have enrolled all patients that were treated in those 20 months at our clinic. Only thos patients who who gave written consent before surgery to participate in this prospective trial were included. All patients who received a intramedullary nail, a hemi- or reverse arthroplasty were excluded. We have treated 90 Consecutive Patients in those 20 months by 5 different trauma surgery. Only those patients who were treated by one surgeon were included because this standardized approach was performed. |
|
|
It seems that no analysis was conducted on complications and fracture types, the surgery difficulty reported by the surgeon, and the bone quality of the patients with complications.
|
We appreciate this comment. We have added the fracture type, difficulty and bone quality into Table 3 for all patients with complication to give more insights.
|
Changes made in Table 3 and lines 282-294. |
|
References seem outdated. Only four out of fifty-three were published in the last five years.
|
We have added 16 more recent references.
|
Changes made in literature. |
|
Please, include biceps surgical treatment among the aims and outcomes of the study.
|
Thank you for this comment. We have added this information into our introduction and material and methods as well as conclusion. |
Changes made lines 69f,, 94ff, 330 ff. |
|
Please, clarify how the sample size was defined. Did the researcher conduct a previous or post-hoc power analysis?
|
We have performed a prospective single center study where all patients treated by this one surgeon where treated in this standardized fashion using a three-hole Philos plate osteosynthesis were included for this clinical and radiographic evaluation. We did not conduct a previous or post-hoc power analysis. |
|
|
Is the surgical technique exactly as described by Minkus et al.? Please, highlight the modifications regarding the original publication, if any.
|
There were no modifications. We have only outlined and summarized the standardized approach for visualizing and educational purposes. |
|
|
Line 88. Please, correct “delta” to deltoid.
|
|
Change made in line 90. |
|
Line 192. Please, confirm that the total number of patients was 41 at this point. When doing the math seems 42.
|
Thank you for this comment. We have corrected this typo.
|
Change made in line 206 |
|
Please add the mean difference and 95% confidence interval when presenting the clinical results.
|
Dear Reviewer, we have added this information into Table 3. |
Changes made in Table 3 |
|
Please, add the fracture types, the surgery difficulty reported by the surgeon, and the bone quality of the patients with complications, and conduct an analysis considering these factors. This is very important, as complications might be present in the fractures, surgeries with higher complexity degrees, or poor bone quality.
|
Dear Reviewer, we have added this information into Table 3.Furthermore we have outlined the analysis of those three intraoperative measure in the results and discussion section. |
Changes made in Table 3 |
|
Please, start the discussion with the main findings of the study. Please, consider restructuring the discussion following these comments: a. a) Complication profiles within techniques might be related not only to the surgical approach (open/miniopen), but also to the fracture type and bone quality. However, the discussion on ORIF did not focus on these parameters. b. b) Lines 306-307. “Nevertheless, a reconstruction should be the goal for young and active patients with proximal humerus fractures”. Please, expand on this idea. The study cohort has a mean age of 63.2 (range 28-92) years, and all the patients were surgically treated with the standardized approach regardless of their age and fracture type. Does that mean all adult patients should undergo ORIF as a primary procedure? Should age only be considered when addressing the biceps tendon? c. c) According to the present study findings, what should be the role of shoulder arthroplasty?
|
Dear Reviewer, we have changed the discussion according to your comments starting with our main findings. We have extended our discussion with regards to fracture type and bone quality which do not correlate with greater complication. As this study did not investigate the results of RSA and did not compare RSA to ORIF, we cannot make any assumptions. However, we all know that RSA is playing an important role in elderly patients for fractures but as we have outlined, age itself is not the important factor. |
Changes made in lines 320-325, 355-357
|
|
The conclusion seems fine but should be revised after addressing the previously mentioned concerns.
|
Thank you. We have added the results with regards to the difficulty and LHB to our conclusion. |
Changes made lines 391ff |
|
Title: Please, consider rewording retention to reduction.
|
We have changed the title according to the comments made by the other reviewer. We have neither included retention nor reduction. |
Changes made in title |
|
The abstract seems fine but should be revised after addressing the previously mentioned concerns.
|
We have shortened the introduction and methods in order to include more relevant information. |
Changes made in abstract |
|
Line 20. “… minimum of 24 months… ”
|
Thank you for your feedback. We have addressed this mistake. |
Changes made in line 19 |
|
References seem outdated. Only four out of fifty-three were published in the last five years. Please, consider its update according to the Editor’s advice.
|
We have added 16 more recent references.
|
Changes made in literature |
|
Table 3. Please, add the fracture types, the surgery difficulty reported by the surgeon, and the bone quality of the patients with complications.
|
Thank you for your feedback. |
Changes made in Table 3 |
Reviewer 2 Report
Respected Authors
My observations on your paper
1- The scientific content is very good
2- Needs improvement in title and abstract section
3- Request you to put the surgical steps and protocol in points and numbers and the same to be labelled in the photographs ,so that it can be easily assessed and understood by readers .
4-Request the authors to change the title from
"A standardized approach for retention and fixation of proximal humeral fractures using a locking plate minimizes primary surgery related complications
TO
A STANDARDIZED OPERATIVE PROTOCOL FOR FIXATION OF PROXIMAL HUMERUS FRACTURES USING LOCKING PLATE TO MINIMIZE SURGERY RELATED COMPLICATIONS .
5-The abstract section needs to be improved further
for example followed up for 24 only is written which should be 24 months
6- Conclusion needs some more points to be added
THANK YOU
Author Response
|
Comments |
Author response |
Changes made |
|
1- The scientific content is very good
|
Thank you very much. |
|
|
2- Needs improvement in title and abstract section
|
We have shortened the introduction and methods in order to include more relevant information. |
Changes made in abstract |
|
3- Request you to put the surgical steps and protocol in points and numbers and the same to be labelled in the photographs ,so that it can be easily assessed and understood by readers .
|
Thank you for this comment. We have added an abbreviation and figure description for the surgical technique. |
Changes made in lines 102-110 |
|
4-Request the authors to change the title from "A standardized approach for retention and fixation of proximal humeral fractures using a locking plate minimizes primary surgery related complications TO A STANDARDIZED OPERATIVE PROTOCOL FOR FIXATION OF PROXIMAL HUMERUS FRACTURES USING LOCKING PLATE TO MINIMIZE SURGERY RELATED COMPLICATIONS .
|
Dear Reviewer, thank you for your feedback. We have changed the title accordingly |
Changes made in title |
|
5-The abstract section needs to be improved further for example followed up for 24 only is written which should be 24 months
|
Thank you for your feedback. We have addressed this mistake. |
Changes made in abstract |
|
6- Conclusion needs some more points to be added
|
Thank you. We have added the results with regards to the difficulty and LHB to our conclusion. |
Changes made lines 391ff |